# Recent Advances in the Behavioral Ecology of European Plethodontid Salamanders

**DOI:** 10.3390/ani13233667

**Published:** 2023-11-27

**Authors:** Andrea Costa, Enrico Lunghi, Giacomo Rosa, Sebastiano Salvidio

**Affiliations:** 1Dipartimento Scienze della Terra dell’Ambiente e della Vita—DISTAV, Università degli Studi di Genova, 16132 Genova, Italy; andrea-costa-@hotmail.it (A.C.); giacomo.rosa@edu.unige.it (G.R.); 2Department of Life Health and Environmental Sciences (MeSVA), Università degli Studi dell’Aquila, 67100 L’Aquila, Italy; enrico.arti@gmail.com; 3Gruppo Speleologico “A. Issel”, Villa Comunale ex Borsino c.p. 21, 16012 Busalla, Italy

**Keywords:** behavior, comparative ethology, experiments, Plethodontidae, salamanders, *Hydromantes*, *Speleomantes*

## Abstract

**Simple Summary:**

Recently, the study of amphibian behavioral ecology has received increased interest from ethologists and evolutionary biologists. In fact, plethodontid salamanders (family Plethodontidae) are often used as model organisms to better understand different aspects of behavioral adaptation. We reviewed the recent scientific literature published on the behavioral ecology of European cave salamanders belonging to the genus *Speleomantes*, to highlight recent advancements and possible future directions for successful research. We found that, in recent years, several aspects of *Speleomantes* behavior were investigated, such as trophic strategies and parental care, while others were neglected, in particular, chemical communication at the intraspecfic level. Finally, we propose European cave salamanders as useful models to understand the gradual adaptation of behaviors that facilitate the permanent colonization of subterranean habitats.

**Abstract:**

There is a recent growing interest in the study of evolutionary and behavioral ecology of amphibians. Among salamanders, Plethodontidae is the most speciose family, with more than 500 species, while in Europe, there are only 8 species, all belonging to the genus *Speleomantes*. European plethodontids recently received increasing attention with regard to the study of their natural history, ecology and behavior; however, the lack of standardized data, especially for the latter, hampers comparative analysis with the species from the New World. We here synthetized the recent advances in *Speleomantes* behavioral ecology, considering as a starting point the comprehensive monography of Lanza and colleagues published in 2006. We identified the behavioral categories that were investigated the most, but we also highlighted knowledge gaps and provided directions for future studies. By reviewing the scientific literature published within the period 2006–2022, we observed a significant increase in the number of published articles on *Speleomantes* behavior, overall obtaining 36 articles. Behavioral studies on *Speleomantes* focused mainly on trophic behavior (42%), and on intraspecific behavior (33%), while studies on pheromonal communication and interspecific behavioral interactions were lacking. In addition, most of the studies were observational (83%), while the experimental method was rarely used. After providing a synthesis of the current knowledge, we suggest some relevant topics that need to be considered in future research on the behavioral ecology of European plethodontids, highlighting the importance of a more integrative approach in which both field observations and planned experiments are used.

## 1. Introduction

The behavioral ecology of amphibians remains largely understudied [1,2], although there is growing interest in the study of ecological processes and adaptive behaviors that shape the evolutionary dynamics of amphibian species and populations [3]. Amphibians represent a highly diverse group that include species with different habitus (from fossorial to exclusively aquatic), different reproductive strategies (oviparous and ovoviviparous) and even individuals that revolutionize their shape through life (larva vs. adult form). According to the intrinsic features of the species, but also to the local environmental conditions, the behavior of amphibians may change dramatically. For examples, in species adopting an *r* reproductive strategy, eggs are produced and laid by females in mass, which are often externally fertilized. On the other hand, in *K*-selected species, parents may spend substantial time taking care of their eggs and newborns, actively fighting against potential predators, driving them to a safer environment, or even feeding their brood [4]. The evolution of parental care in amphibians indeed seems to have played an important role for their colonization of different habitats. Looking at bi-phasic species, the behaviors of aquatic larvae can be completely different from those adopted by terrestrial adults. It is, therefore, quite complicated to standardize and compare amphibians’ behavior, unless specific groups with a similar ecology and life traits are considered.

Among Urodela, the family Plethodontidae (Gray, 1850) is the most speciose [4]. This family comprises more than 500 species out of 800 worldwide salamander and newt species, of which the overwhelming majority is found in Northern, Central and Southern America, while there are just 8 species in Europe and only 1 in Asia [4,5,6]. Plethodontids are characterized by the absence of lungs at the adult stage, and by the presence of naso-labial grooves that connect the upper lip to the external naris [4]. The main function of these peculiar structures is to canalize chemicals towards the olfactory structures [7,8].The wide diversity and the high local abundances that New World Plethodontids can reach make them relevant nodes within local ecological webs, where functional guilds are composed of many different interacting species [9,10,11,12]. Indeed, epigean plethodontids are predators which often occupy an intermediate position in the local food web, meaning that they hold the critical role of being prey and predators at the same time. Many species of New World plethodontids occur sympatrically, a condition that allowed several studies on species habitat selection and competition to be performed. Plethodontid salamanders have been even used as proxy for biodiversity assessments and monitoring in North American forests, cases that further highlight the importance of this group of amphibians and the need to deepen our knowledge on their biology and life traits [3].

On the other hand, the discovery of the Old World plethodontid species is relatively recent, and studies on their ecology and behavior lagged behind compared to their American relatives. Some not exclusive causes may contribute to this knowledge gap. As far as we know, Asian and European plethodontid species usually do not occur in sympatry, preventing the completion of studies on competitive behaviors of species living sympatrically. To this purpose, syntopy between two European plethodontid species has been artificially created, but such unnatural conditions do not provide any substantial information for advancing our knowledge on their behavior. Furthermore, although eight species of European plethodontids are currently known, they have been historically considered equivalent in their biology and ecology, meaning that single-species studies were often translated to the whole genus (see [13] and references therein). Indeed, only recently has more emphasis been given on evaluating interspecific (or even intraspecific) divergences among European plethodontid species, emphasis that strongly contributed in raising the number of studies performed. Furthermore, European plethodontids usually occur in subterranean environments, habitats where prolonged studies and monitoring activities are more challenging compared to surface ones.

American plethodontids became one of the main amphibian model systems used to better understand ecological questions in amphibians such as interspecific competition, predation and hybridization. Starting from field observations, classical ecological manipulative experiments were carried out, initially in natural woodland habitats [14,15,16]. However, in natural settings, there are many external uncontrollable factors that may influence the observed outcomes. For these reasons, plethodontids are often tested in laboratories to facilitate the avoidance of external confounding factors, while simultaneously controlling biological variables such as age, sex or reproductive status [17,18,19].

In this context, the current understanding of the behavioral ecology of European plethodontids, known as European cave salamanders, remains poorly developed in comparison to American plethodontids. The eight European species belong to the genus *Speleomantes* Dubois, 1984, although they are sometimes referred to as *Hydromantes* [5,20,21]. However, the use of *Speleomantes* better highlights the phylogenetic independence of European plethodontids, which form a well-defined monophyletic group with a large genetic distance from the five Californian species of *Hydromantes* [22,23,24]. All European cave salamanders are fully terrestrial and usually inhabit habitats where specific microclimatic conditions (e.g., high humidity and relatively low temperature variations) occur, such as humid crevices, forest floor environments and natural or artificial subterranean habitats [13,25,26]. Historically, behavioral studies on European plethodontids were sporadic because zoologists were focusing mainly on species description at the morphological and genetical level [25,27].

The monography dedicated to the taxonomy, biogeography and ecology of European cave salamanders written by Lanza et al. [13] constitutes the main reference concerning the natural history, ecology and behavior of the genus *Speleomantes*. This monograph lists under the chapter “Ethology” four sub-sections: “Feeding behaviour”, “Activity, habitat use and displacement”, “Antipredatory adaptations” and “Communication”. Concerning “Feeding behaviour”, most of the studies dealt with the anatomical structure of the eye [28] and of the tongue [29,30]. This sub-section summarizes the extraordinary tongue protrusion capability of *Speleomantes* and the ability to capture static prey in complete darkness [31,32]. Finally, the trophic strategy of *Speleomantes* is described as an “ambush strategy”, although this assertion seems based more on sporadic observations than on robust evidence [13]. The second sub-section, “Activity, habitat use and displacement”, relates exclusively to the species’ auto-ecological requirements, such as habitat features and seasonal activity, with little connection to behavior ecology *sensu* [3]. The third sub-section, “Antipredatory adaptations”, describes the presence of yellow-reddish or ochre dorsal colorations and the production of toxic skin secretion observed in many *Speleomantes* individuals and populations [13]. Finally, in the fourth sub-section, “Communication”, the presence of chemical intraspecific interactions was inferred from anatomical and histological studies on mating glands, but without providing any experimental evidence of their use during courtship and mating. In any case, the typical “nose tapping” behavior of plethodontids was reported in *S. strinatii*, suggesting that this specific behavior related to chemical communication is present [33]. Consequently, the relative paucity of the literature on European cave salamanders’ behavior does not allow robust comparative studies with American plethodontids [3,34,35].

The principal goal of this work is to provide a synthesis of the current knowledge on *Speleomantes* behavioral traits, and to stimulate new research aimed on this topic to produce a substantial amount of research that enables comparative studies with New World plethodontid species. Furthermore, we aim to propose this genus as a model to test eco-evolutionary hypotheses. For example, *Speleomantes* are facultative cave species that use subterranean environments to avoid unsuitable climatic conditions [36]. The microclimatic conditions offered by subterranean environments promote high efficiency for *Speleomantes* cutaneous respiration; therefore, European cave salamanders often establish stable populations in these habitats. The colonization of subterranean environments by epigean populations likely begins with a series of behavioral adjustments, allowing individuals to better exploit the new environment [37]. In this framework, *Speleomantes* could be considered as troglophiles *sensu* [38] and should be useful models to study the dynamics of meta-populations connecting subterranean and epigean ecosystems. 

## 2. Materials and Methods

Peer-reviewed full papers, including proceedings from national meetings, published from 2006 to 2022, were considered in this study. Papers dealing exclusively, or giving relevant information, on European plethodontids were selected from online databases (i.e., Scopus and Clarivate Web of Sciences), searching for “Speleomantes” or “Hydromantes”. General articles just citing *Speleomantes* (or *Hydromantes*), faunistic lists, atlases and abstracts from scientific meetings were excluded. Papers dealing with the American *Hydromantes* species were discarded. Therefore, only European plethodontid were taken into consideration and analyzed.

Initially, we evaluated the existence of a recent trend in publications concerning the European genus *Speleomantes*, both in their absolute number and in relation to the continuous growth of global scientific production [39]. To obtain this latter information, we calculated the ratio between the number of papers dealing with *Speleomantes* or *Hydromantes,* but referring only to European species, and the number of papers with the word “amphibians” in the title, abstract and keywords retrieved each year from Scopus (accessed 10 August 2023). Temporal trends were analyzed using the Mann–Kendall non-parametric test [40], while we assessed an equal article distribution among *Speleomantes* species using a χ^2^ test for homogeneity [41], counting multi-species papers as one entry for each species.

In addition, we grouped European plethodontid studies according to their general subject, using as inspiration the chapter titles of the book *Behavioral ecology of the Eastern Red-back Salamander—50 years of research* by Jaeger et al. [3] as a benchmark. This choice was made for two reasons: (i) to use a clear operational definition of “behavioural ecology” that was already applied to salamanders by a leading research group for over 40 years, and (ii) to allow a better comparison with the research subjects on *Plethodon cinereus,* which is the most studied North American plethodontid species. However, we made some modifications or additions to better describe the studies published on European cave salamanders (see Table 1). For example, the first chapter “Interspecific competition between *P. cinereus* and *P. shenandoah*” was too specific and could not be applied to *Speleomantes*, while papers on courtship and parental behavior (topics not treated in [3]) were added to the category “Intraspecific social behaviour”. In order to better understand the types of studies within each behavioral category, we distinguished the two sub-categories “experimental” and “observational”. Papers were considered “experimental”, if they were carried out in controlled environments (i.e., in laboratory enclosures and terraria) or in the field with some type of intervention planned to discriminate among alternative hypotheses. In contrast, studies were considered “observational”, if they were based on data obtained from field populations without any previous manipulative intervention.

## 3. Results

We retrieved 26,180 peer-reviewed papers published from 2006 to 2022 with the word “amphibians” from the Scopus database. During the same time frame, 106 full papers dealing with European plethodontids were published (Appendix A). During the study period, there was a positive trend in both the general scientific output on “amphibians” and for European plethodontids, expressed as the absolute number of papers published per year (Mann–Kendall trend test, Z = 3.831, *p* = 0.0001, and Z = 3.253, *p* = 0.001, for amphibian and *Speleomantes* papers, respectively). The relative proportion of *Speleomantes* papers showed a positive trend, even considering the general increase in overall papers published on amphibians (Figure 1; Mann–Kendall trend test, *Z* = 2.595, *p* = 0.009).

Among these European plethodontid peer-reviewed papers, 34% (36/106) could be classified as broadly dealing with behavioral ecology as defined in this study (Figure 1), while the complete list of citations is given in Appendix A. Behavioral papers mainly focused on two subject categories that accounted for 75% of the retrieved articles: “Foraging tactics” (42%, 15/36) and “Intraspecific social behaviour” (33%, 12/36; Figure 2). In the last category, six articles dealt with parental care in the genus *Speleomantes*, while the remaining articles focused on other intraspecific interactions. There were no studies on “Pheromonal glands and pheromonal communication” and on “Interspecific behavioural interactions”. Most studies were observational (83%, 30/36), while the remaining studies used an experimental approach in controlled or natural environments. During the study period, the lowest number of research articles was dedicated to *S. ambrosii* (*n* = 4) and the highest to *S. strinatii* (*n* = 15; Table 1), with a relatively homogeneous distribution among the eight *Speleomantes* species (χ^2^ = 13.271, df = 7, *p* = 0.066).

## 4. Discussion

Despite the increasing number of peer-reviewed papers published since the review of Lanza et al. [13], the behavioral ecology of European plethodontids belonging to the genus *Speleomantes* remains poorly known.

From our analysis, it is clear that recent research interests were focusing on three main topics in particular, related to foraging tactics, intraspecific social behavior (but the majority of papers were dealing with clutch guarding and parental care rather than on interactions among free-living conspecifics) and with predators or parasites. Foraging ecology is very relevant when assessing the role of salamanders as top predators on invertebrate communities that inhabit the superficial soil stratum [3]. Therefore, the dietary habits and feeding behavior of all eight species of *Speleomantes* were investigated at least once (Appendix A). Researchers mostly explored the trophic spectrum of *Speleomantes* with the analysis of stomach contents, identifying diet composition in multiple conspecific populations and assessing how seasonality contributes to defining the type and amount of consumed prey [42,43]. In some instances, assessments of individuals’ food specialization were also performed [19,44]. On the other hand, many other aspects of *Speleomantes* foraging behavior are still unexplored or just hypothesized [45]. For example, we still do not know which foraging strategies these species adopt, or where they forage the most. Furthermore, do *Speleomantes* show prey preferences? Studies of gut contents were usually not supported by analyses aiming to quantify the local trophic supply, but see [46,47], making research topics related to individuals’ prey preferences virtually unexplored in many species. To date, all trophic ecology studies were performed using a stomach flushing technique on live individuals, a relatively simple technique that has been often performed in the field on different species of salamanders [46], while stable isotopes or DNA barcoding are rarely used [48].

Most of the studies regarding predation and parasitism on *Speleomantes* deal with the observations of these events in both epigean and ipogean environments. Consequently, the knowledge regarding this topic is still lacking (e.g., potential predators and predation avoidance) and more experimental studies should be carried out to explore the subject.

The second most explored research topic was related to reproductive behavior and parental care. Considering the cryptic behavior of *Speleomantes*, especially when they reproduce, the discovery of some nests in the wild is of high importance to comprehend their reproductive behavior, although it is limited to some species and mostly concentrated in the spring–summer period [36,49]. Indeed, most of the available information identifies the beginning of spring as the time in which females usually lay eggs, which are then protected until they hatch at the end of summer [50]. Different is the case of parental care. Using a facility provided by a semi-natural laboratory, some researchers were able to study female parental care in *S. strinatii*. Researchers observed an active protection of the mothers of both their eggs and newborns [51]. Future studies aiming to deepen research on this topic should include more species and (hopefully) include wild observations, as individuals may change their behavior when translocated into different environments [52].

The lack of interspecific behavioral studies may be due to the allopatric distribution characterizing the genus *Speleomantes.* This condition, to the best of our knowledge, does not occur in only two narrow contact zones of mainland Italy, where, inter alia, hybridization between species is present [53,54]. In the past, a couple of experiments involving the creation of artificial sympatry of two mainland *Speleomantes* species were performed; their aim was to study their habitat selection and competition [55]. However, data from such unnatural conditions are useless for this review; therefore, those studies were not considered. In addition, only two other terrestrial salamanders are found in sympatry with one *Speleomantes* species at a time, i.e., the fire salamander *Salamandra salamandra* and the Northern spectacled salamander *Salamandrina perspicillata* [13]. Such co-occurrence is mostly realized in epigean environments (e.g., in forested areas), although both species can occasionally exploit subterranean environments [56,57]. A few studies performed on this system provided the first information related to the potential mechanisms (e.g., temporal and trophic niche partitioning) that these species adopt to avoid competition [58,59]. However, more natural observations and experimental tests are needed to shed light on this topic.

The behavioral trait for which our knowledge is the poorest is *Speleomantes* communication and social interactions at an intraspecific level. We still have uncertainty on the nature of intraspecific interactions between individuals. Some authors highlighted the potential occurrence of agonistic interactions between age classes [60] within subterranean environments, while some others found no evidence of such behavior [61]. The single study performed on surface population does not report any evidence of agonistic interactions [62].

Finally, our review highlighted the relative low number of experimental studies published on European plethodontids. New World plethodontids are currently used as laboratory models for experimental studies on comparative and adaptive behavior [3]. Characteristics promoting the use of plethodontids as model species are their easy maintenance, survival and breeding in animal research facilities [3]. In controlled environments, selected individuals can be exposed to experimental conditions to test for multiple hypotheses. In fact, the experimental hypothesis-testing approach seems very informative in the study of behavioral ecology, since it allows researchers to infer causality [3,17,18]. Although, caveats should be always considered when extrapolating results observed in simplified, and possibly stressing, settings compared to natural habitats [19]. This type of approach was performed mostly on the continental species *S. strinatii*, with only one exception concerning *S. italicus* (see Appendix A). However, similarly to many New World plethodontids, *Speleomantes* are medium-sized terrestrial salamanders that can be easily hosted in the laboratory within terraria or small containers for relatively long periods of time. However, reproducing a suitable microhabitat for these species may be not trivial. In fact, due to their strictly cutaneous respiration, European plethodontids are highly dependent on microclimatic conditions, which should be characterized by high air relative humidity, and air temperatures usually lower than 18 °C [26]. Therefore, maintaining these environmental features within such ranges would assure animal welfare. These environmental conditions can be reproduced only in laboratory cold rooms or in natural or artificial subterranean habitats, thus limiting the experimental approach, as we, in fact, observed. Based on our analysis of the recent published research on European cave salamanders, we propose a preliminary list of possible topics for future research (Table 2).

## 5. Future Directions

Our analysis also highlighted the scarcity of comparative studies, where the behavioral adaptations of the *Speleomantes* species are framed in an evolutionary perspective. These studies are useful to determine if variation in behavioral traits is correlated with the phylogenetic relatedness or with local ecological conditions experienced by the different species [63]. This comparative approach should be one of the most fruitful lines of research, because the phylogenetic relationships among the eight European cave salamander species have been relatively well investigated and quantified using different approaches [64,65].

Another relevant issue concerns understanding the evolutionary processes that allowed European cave salamanders to colonize subterranean environments [66]. This, in fact, is one of the hottest topics in evolutionary biology and for biospeleologists; although, it is not always easy to accomplish [52]. Most of the effort focused on understanding the adaptive traits characterizing the most adapted subterranean species (i.e., troglobites), while little attention was given to facultative cave species such as troglophiles and trogloxenes [38,67]. Specific traits characterizing troglobites (e.g., anophtalmia, lack of pigmentation, high longevity) are peculiar and very attractive for investigative purposes; indeed, researchers very often compare subterranean species with such characters with their genetically close-related epigean species [68,69,70]. However, the appearance (or disappearance) of a specific character can be a long stepwise process, where species not completely adapted to subterranean environments may provide key information on the intermediate processes from one end to the other. Improving behavioral studies on *Speleomantes* can be very useful to understanding such evolutionary processes, as behavioral adjustments likely take place at the very beginning of the colonization of subterranean environments [37,71]. Using *Speleomantes* as model species provides the (very rare) possibility to employ in experiments both epigean and subterranean conspecific populations, a condition that will help in producing more robust inferences and more clear results as it reduces potential divergences due to different life histories. A similar approach has already been successfully used by comparing the surface-dwelling and the cave-adapted populations of the fish *Astyanax mexicanum* [72,73] and those of the Pyrenean brook salamander *Calotriton asper*. The adults of this latter salamander are terrestrial but mate and lay eggs in streams and produce aquatic larvae. In this species, both epigean and hypogean populations are known [74], and many studies aiming to assess behavioral divergences between surface and subterranean populations have been performed. For example, to test the hypothesis that cave-adapted individuals evolved increased food-finding abilities, the prey detection performances of surface and subterranean populations were compared [72]. This study showed that epigean Pyrenean brook salamanders possess a predisposition to successfully forage at night and that these salamanders improve their ability to forage in total darkness when experimentally maintained in cave conditions for long periods of time. Therefore, this plasticity in foraging behavior may facilitate the permanent colonization of subterranean habitats, in which food is limited, and light is reduced or completely absent [72].

In conclusion, our analysis showed the many knowledge gaps hampering a proper understanding of the behavioral ecology of *Speleomantes* species. Starting from the many observational studies published in recent years, an experimental approach should be promoted to better understand the behavioral adaptations described for the different species. For example, under controlled conditions, such as when using terraria or mesocosms, the existence and relative strength of intra- and interspecific chemical communication could be successfully tested. Finally, a more evolutionary and comparative approach should also be encouraged, as is the case of Lunghi et al., (2018) [50], which is the only study comparing the reproductive behaviors of all eight species of European cave salamanders.

## Figures and Tables

**Figure 1 animals-13-03667-f001:**
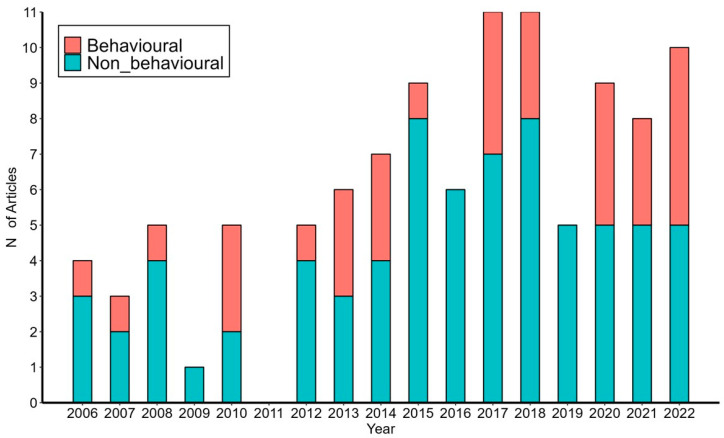
*Speleomantes* papers published from 2006 to 2022. Red subgroups correspond to articles dealing with the behavioral ecology of *Speleomantes*, while blue ones cover the other topics.

**Figure 2 animals-13-03667-f002:**
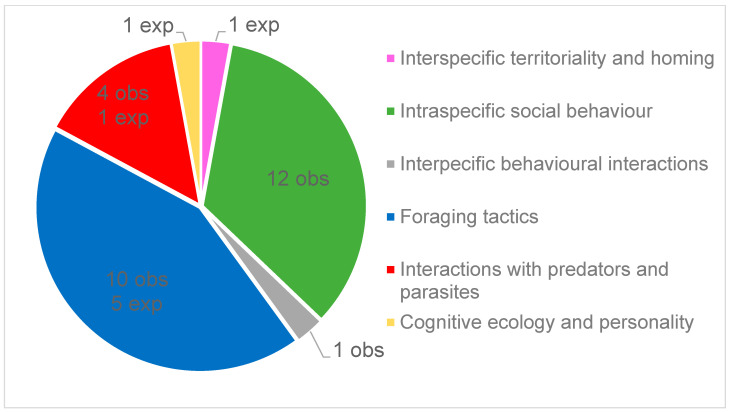
Distribution of behavioral ecology subjects of papers published on the eight *Speleomantes* species, together with the detail of how many of these studies are observational (obs) or experimental (exp).

**Table 1 animals-13-03667-t001:** Behavioral ecology categories and related predominant species considered in this study (modified from [3]).

Behavioral Ecology Category According to [3]	Behavioral Ecology Category Used in This Study	Most studied Speleomantes Species for Each Category
Interspecific territoriality	Interspecific territoriality and homing	*S. strinatii*
Foraging tactics	Foraging tactics	*S. italicus*
Pheromonal glands and pheromonal communication	Pheromonal glands and pheromonal communication	/
Interspecific behavioral interactions	Interspecific behavioral interactions	*S. strinatii*
Intraspecific social behavior	Intraspecific social behavior, courtship, mating, and parental care	*S. strinatii*
Interactions with predators	Interactions with predators and parasites	*S. italicus*
Cognitive ecology	Cognitive ecology and personality	*S. strinatii*

**Table 2 animals-13-03667-t002:** Suggested topics for future research on European cave salamanders species.

Behavioral Category	Behavioral Ecology Question	Suggested Focal Species	Suggested Experimental Approach
Intraspecific social behavior, courtship, mating and parental care; Pheromonal glands and pheromonal communication	Do individuals communicate through tactile, visual or chemical signals?	All species	Experiments/observations in controlled conditions
Intraspecific social behavior, courtship, mating, and parental care; Pheromonal glands and pheromonal communication	How is mate selection made?	All species	Experiments/observations in controlled conditions
Intraspecific social behavior, courtship, mating, and parental care	How different species/populations adjust their behavior depending on habitat features?	All species	Experiments/observations in controlled conditions or in the wild
Intraspecific social behavior, courtship, mating, and parental care	Is the behavior of hybrid individuals related to that of a parent?	*S. ambrosii*, *S. italicus*, *S. strinatii* and their hybrids	Experiments in controlled conditions of genetically identified individuals
Cognitive ecology and personality	Are space use, displacement patterns and overall fitness linked to individual behavioral traits?	All species	Capture-mark-recapture study in the wild
Cognitive ecology and personality	How individuals modify their behavior after environmental or social experiences?	All species	Controlled conditions in outdoor mesocosms
Interspecific territoriality	How interspecific coexistence shapes resource use or activity patterns?	*S. ambrosii*; *S. italicus*; *S. strinatii*	Controlled conditions as laboratory settings or outdoor mesocosms

## Data Availability

The data presented in this study are available in the Appendix A.

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
