# Peer review of "Recent Advances in the Behavioral Ecology of European Plethodontid Salamanders"

_animals, 2023, doi:10.3390/ani13233667_

Round 1
Reviewer 1 Report
Comments and Suggestions for Authors
Dear Andrea Costa, Enrico Lunghi, Giacomo Rosa, and Sebastiano Salvidio! I have read your manuscript “Recent advances in the behavioural ecology of European plethodontid salamanders”. I agree that 36 primary research papers on behavioural ecology of the genus Speleomantes is a nice achievement for a good 15-year time span and that the time is ripe to review and synthesize the knowledge scattered in this primary literature. Especially, if the authors wish to push this field of research further as is evident from the text.
While reading the manuscript I did not detect any major scientific content-wise issues. However, I see no benefit to organise your review in the classic IMRaD format. Reviews in general rarely follow IMRaD if they do not include some extensive statistical analyses. In my opinion, your review is not appropriate for this format and will in fact benefit if you choose to significantly restructure and reorganize it. I see two main reasons for this.
First, statistical analyses of the literature scan performed are not really needed as they add nothing content-wise. Also, statistics are not comprehensively explained, and I had difficulties or even could not understand certain parts of the Results section, including Figure 1 (see minor comments below). I suggest simplifying things and keep only the basic information. Reorganize the current Methods and Results sections into one, much shorter section entitled e.g., Literature scan, that will follow the introduction. The basic info within this section should be: 106 papers on Speleomantes were published between 2006 and 2022 (btw, congratulations!) as discovered via a Scopus and CWS literature scan; 36 (34%) of these focus on behavioural ecology; I would show this on Figure 1 (histograms with no. of papers per year, but use stacked histograms to differentiate beh eco papers from the rest; no trendline); this will give all relevant info, I think no need to show “all amphibian” papers as background or do any statistics. Then add the stuff on how many of the 36 papers focus on each beh eco category (according to Jaeger et al. 2016), are observational vs. experimental, and which species predominate (you already have all of this). Remove Figure 2 (just say the numbers in text) and the chi2 test (not clear why and how you did it, and its results not clearly interpreted, and not needed at all).
Second, currently there is no real summary or synthesis of the results and knowledge gained from the 36 focal papers that the reader could benefit from without reading the primary literature. Thus, I suggest that you devote the central part of the paper to a summary&synthesis organized by behavioural categories sensu Jaeger et al. 2016. In each subsection provide a summary and synthesis of current state of knowledge learned from these 36 papers. You could use some of the text from the discussion for this but should add novel text too. This way the reader will also realize for itself which topics are well explored, which are not, and where are the main knowledge gaps.
The last part of the paper should be called something like “Future prospects” and should be built around your Table 2 and explain and comment it. Here also include the last two paragraphs of the current discussion.
Lastly, you propose Speleomantes as a model system to investigate colonisation of subterranean environments (in introduction and in discussion). I agree, it is a valuable system and a great opportunity. But there are others, even among European urodelans: Salamandra salamandra and Calotriton asper regularly enter caves while Proteus anguinus is bounded to cave life. I think the reader will benefit if you include a short comparison of Speleomantes to these taxa (add additional urodelans if you are aware of their associations with subterranean habitats). What is the benefit of Speleomantes against the others? I think your review will be useful also to compare to these other European urodelans, not only the American relatives of Speleomantes. To do so, please add a short summary of what is known of their behavioural ecology and support it with the most relevant and/or recent references (e.g.: Guillaume 2022, Premate et al. 2022, see below for full references). You could do this either in in the last part of the introduction or discussion / future prospects. This will give the reader a better understanding of how valuable your vertebrate model system is for understanding the colonisation of subterranean habitats.
Guillaume O (2022) Surface newt Calotriton asper acclimation to cave conditions improved their foraging ability in darkness. Front. Ecol. Evol. 10:1057023. doi: 10.3389/fevo.2022.1057023
Premate E, Fišer Ž, Kuralt Ž, Pekolj A, Trajbarič T, Milavc E, Hanc Ž, Kostanjšek R (2022) Behavioral observations of the olm (Proteus anguinus) in a karst spring via direct observations and camera trapping. Subterranean Biology 44: 69-83. https://doi.org/10.3897/subtbiol.44.87295
MINOR COMMENTS (if you chose to reorganise your paper according to my suggestion, some of these minor comments might be obsolete and please ignore them):
Introduction
Line 35-36: wrong figure numbering, or better still just delete the reference to the figure
Line 74: natural or artificial subterranean habitats
Line 75: consider starting a new paragraph here
Methods
Line 122: Speleomantes in italic
Line 123: consider starting a new paragraph here
Line 130-131: Speleomantes in italic
Line 131: consider starting a new paragraph here
Line 137: Plethodon cinereus in italic
Line 152: replace topics by categories
Line 155: not clear why the asterisk is needed. Either delete or explain in annotation.
Results
Line 160-161: The table in supplement should have a unique title, perhaps Table S1, to not confuse it with other tables. Like you wrote in Line 177.
Figure 1: not clear to me what is actually plotted; please name the y-axis. Y-values seem high, if this is the ratio of Speleomantes vs amphibian papers through time? Also what do the columns represent, and what is the trendline? How did you plot it if MK test in non-parametric? Did you use linear regression as well? The part dealing with trend analysis of Speleomantes literature relative to amphibian literature is confusing.
Line 184: no need for a new paragraph here, join with the previous one
Line 185-186: not clear what you are saying or how to interpret the statistical results.
Line 185: S. Strinatii in italic
Line 186: Here you wrote Chi-quare, while in line 131 above you wrote X2. Please be consistent.
Line 189: renumber to Figure 2 not 1
Line 190: how you treated with multispecies papers should be explained already in the methods.
Line 195-198: This text is a remnant of some formatting instructions. Please delete.
Discussion
Line 203&205: Speleomantes in italic
Line 204: Figure 1 does not show what you claim it does.
Line 214: consider starting a new paragraph here
Line 202: replace “their” by “they”, or replace “their hatch” by “hatching”
Line 227-229: Please rephrase this sentence. First you say that species are allopatric, than you mention two contact zones (peripatry). It is confusing.
Line 230&241: Speleomantes in italic
Line 239: please say directly which mechanisms are these
Line 245: replace “did have” by “found”
Line 249&250: not clear what you mean. Rephrase.
Line 250: S. strinatii and S. italicus in italic
Line 260: delete “reasoned”
Table 2: First column is redundant as the same information is given in the third column (beh eco question). Please remove it but potentially use key words from it and include them in the third column questions. Make sure that the behavioural categories in the second column are the same as in Table 1, second column. Or is there any special reason that you use Jaeger’s naming and not your updated names for the categories? Are you sure that the second row (personality question) should not be listed under the category Cognitive ecology and personality? Also make sure that in column three you phrase the text as question(s) to match the column title. Order rows by value in behavioural categories column. No reason to interrupt the questions for “Intraspecific social behaviour, courtship, mating, and parental cares” by other categories/questions as now. Finally, I expected at least one question identified for each of the behavioural categories. Now, one category has many questions, and some have none. The reason for this is not that the lacking categories have been well explored before. It feels unbalanced, and as if you have a special interest in something. This is ok, but you are giving a review, and should try to be objective and inclusive.
Line 287: delete “in which are lacking”
Line 307: add “salamanders”.
Supplementary information
Please explain in the table title that papers marked in yellow focus on some behavioural ecology topic. This is not clear now at first sight.
Comments on the Quality of English LanguageI have not given a thorough review or comments on language or style as I suggest a major reorganisation of the manuscript.
Author Response
Dear Editor,We apologize for the delay in uploading the revised manuscript. The Reviewers requested many changes and our comments took a a long time. Furthermore, the tables and figures have also been modified. We have responded to the comments of the two Reviewers' separately in the attached documents.

Reviewer 2 Report
Comments and Suggestions for Authors
The authors review literature from 2006-2022 on the behavioral ecology of European cave salamanders. The authors find that there are many gaps still in our understanding of the behavior of cave salamanders, especially in regards to intraspecific interactions. In addition, most studies were observational, and not experimental in nature.
While I think that a paper such as this would be a valuable contribution to the literature. The current manuscript needs substantial revisions in order to be considered for publication. Besides a restructuring and augmentation of the introduction and discussion, the methods of the paper also need to be clarified and made more robust so that the results represent an unbiased and replicable study.
I have broken down my comments by manuscript section to make them clearer.
Intro
· The organization and content of the intro could be improved to set up the paper. As it is currently written, I don’t think that the introduction sets the stage for the rest of the paper or the knowledge gap that it is filling as well as it could.
· In the opening paragraph, the content starting on Line 38 does not match the first, opening sentences of the paper. According to the initial sentences, this paragraph should be about what is known/lacking/recent advances in amphibian behavioral ecology, not specifics about Plethodontids. I suggest the authors have an independent first paragraph that addresses and summarizes eco-evo behavioral studies of amphibians before diving into Plethodontids.
· Much like the first paragraph, the topic sentence and the content of the second paragraph do not align. The idea that Plethodontids are a model organism in ecology is distinct from the idea that they reach high densities. Consider making a separate paragraph that summarizes what is known about the behavioral ecology of New World Plethodontids. This would also make your argument that little is known about European cave salamanders more robust.
· I'm a bit confused as to why so much text is spent on the Lanza et al. review. I think that that paper should be presented as the most-recently published review on this group, but that readers should be directed to the original publication to find out more details about what it says, rather than summarizing it here. The authors could highlight the gaps in the behavioral ecology literature identified in this previous review to highlight the need for their review.
· The idea that this group could be a model system should be presented in the discussion, not in the introduction. The reader should be thoroughly introduced to them before stating that they should be a model system.
· Specifics:
o Lines 35-36 - A figure reference in the first sentence of the introduction is odd - this should be the topic sentence of the paragraph and thus not contain any specific figure references. What is more, the first reference is to Figure 2A -- Figures should be referenced in the order that they are used. There is also no Figure 2A, apparently.
o Lines 46-48 – I am not sure what the authors intend by the end of this sentence - is it a definition of an ecological web? Could the authors clarify?
o Lines 52-63 - This section promoting controlled experiments seems odd here - this section seems better suited to the discussion. At this point, the authors should be summarizing what is known, not touting the use of experimental approaches over observational studies
o Line 77 – I would start a new paragraph after the sentence that ends early on this line. Otherwise, this paragraph becomes unruly.
o Lines 98-100 – This a dramatic way to discuss the relative amounts of literature between New World and European cave salamanders and, as such, it is difficult to follow. I would just highlight the relative paucity of literature and state that robust comparative studies with American plethodontids cannot be completed.
o Line 102 – “Current knowledge” suggests that you will be reviewing all published literature on the behavioral ecology of this genus when, in reality, you are just look at 2006-2022. I would rephrase to highlight that you are just reviewing literature since the Lanza et al. monograph was published.
Methods
· The methods used in this study do not align with those commonly used for literature reviews. There should be clearly defined search terms used to find references and references should be pulled from publicly-available databases (not authors personal repositories) in order for this study to be repeatable.
· The papers used in the review should be retrieved using the same methods as the number of papers with “amphibian” in the title, otherwise these two values/trends are not comparable.
· This section needs to be restructured. There should be separate paragraphs/subsections defining:
o 1) How the articles were retrieved
o 2) what analyses/data categorizations were done with separate sections for each analysis
· Lines 131-149 - I'm not sure that the level of detail included here is necessary. It would be sufficient to say that you used the titles of that source as inspiration, making modifications as needed to make the categories more inclusive.
· The Table included in this section is not necessary. Subtle differences between behavioral categories between this review and a previous review are not vital enough to warrant a table.
· Specifics:
o Lines 130-131 – Genus should be italicized
o Line 134 – Inconsistent reference formatting
o Line 137 – Scientific name should be italicized
o Lines 146-148 – This specific example is not needed.
Results
· Line 164 - How did you get this relative proportion value? What is this a relative proportion of?
· Both Figure 1 and 2 – These figures are missing y-axis labels
· Line 185 – Scientific name should be italicized.
· Figure 2 – is labeled as Figure 1. There is also a “caption” for Figure 2 that seems to be copied in from the author instructions as it is not a true caption for the figure. (Lines 195-198)
Discussion
· This section would make more sense if broken down into subsections by behavioral category – e.g., Foraging Behavior. This would make this section easier to follow and allow for more thorough discussion of the patterns that the authors observed and allow for the authors to highlight gaps and future directions
· Table 2, while a nice thought, would be better suited for a Supplement. The discussion should focus on summarizing the findings; thoughts and suggestions at this level of detail don’t belong here.
· Specifics
o Line 203 – Genus not italicized
o Lines 203-204 – Figure 1 does not support the claim made in this sentence
o Lines 228-229 – If the species do not broadly overlap, as this sentence implies, how can they be allopatric in only two locations?
o Line 230 – Genus not italicized
o Lines 231-232 - How are these data useless? How is behavior after translocation acceptable, but not this?
o Line 235 – Pages not necessary to accompany reference
o Lines 237-240 - Some additional details should be provided on the studies presented here. Otherwise it is unclear how the previous studies were deficient.
o Line 241 – Genus not italicized
o Line 241-242 - This topic sentence should highlight that you are specifically discussing intraspecific interactions
o Line 250 – Two species names not italicized
o Line 271 - Was the number of comparative studies provided? I did not see this discussed in the methods or results section
o Line 293 – “preciousness” is an odd word choice that is, frankly, too value-laden to use here.
o Table 2
§ If one is interested in animal personality, wild populations would make this exceedingly difficult as it would rely on very high recapture rates
§ Hybrids has not yet been mentioned in this paper. Should be addressed in the early background on the group
Conclusions
· The authors tout the use of experimental studies, but also claim that such studies were “useless” for their analysis because such studies are “unrealistic”. So, it is unclear where the authors stand on experimental vs. observational studies and the utility of each to advance this field.
· Line 307 – appears to be missing a word at the end of this sentence.
Comments on the Quality of English Language· There are many sentences that are joined with a semicolon that would be easier to read if made into separate sentences.
· Some examples of grammatical errors:
o There is a period in the middle of the opening sentence
o Line 38 - Gray, 1850 should be in parentheses
o Line 45-46 - “Syntopically” is not the correct word to use here. It should be “sympatrically”
o Line 59 – Specific pages are not required in text with references
o Line 215 – “parental cares”
Author Response
Dear Editor,
please find attache our replies to Reviewer 2.

Round 2
Reviewer 1 Report
Comments and Suggestions for Authors
Dear authors,
Thank you for an appropriate revision of you manuscript. In my opinion the revised version is much better and I have only one major comment left, which will however not take much of your time to incorporate into the text. Please consider it carefully. I also provide a list of minor comments.
Major comment:
Line 307-313: I completely agree with you here, but also think that you should mention another case for a complete record. One common way in comparative biospeleobiology is to contrast a subterranean species to a closely related epigean species. Often these comparisons offer only limited insight as usually species used are not so closely related after all (for example Proteus and Calotriton that you used as a reference here). The other common approach is to contrast subterranean populations with epigean populations in species like Astyanax mexicanus, Asellus aquaticus, Gammarus minus, etc. These are very closely related but also their populations are very distinct. This might be the best approach. Finally, come your model organisms (European cave salamanders and similar), whose subterranean and epigean populations do not differ so markedly as in the previously mentioned species. So, I suggest adding a sentence or two, mentioning the above model organisms and providing appropriate reference. For these I suggest the latest reviews, chapter in the recent Groundwater Ecology and Evolution monograph.
https://doi.org/10.1016/B978-0-12-819119-4.00004-4
https://doi.org/10.1016/B978-0-12-819119-4.00016-0
https://doi.org/10.1016/B978-0-12-819119-4.15007-3
Minor comments:
Throughout the manuscript genus names are often not written in italic. This is especially apparent in the simple summary, abstract and future directions. Please see to it.
Line 57: change Amphibians to amphibians
Line 71: check formatting of citations
Line 81: replace - after gap by a full stop.
Line 82: check (but see), something is missing
Line 88: not clear which paper by Lanza you are referring to as there is no year of publication
Line 90: check formatting of (e.g.), something is missing?
Line 136: delete redundant space between literature and on
Line 137: delete redundant space between with and American
Line 193: usually stats labels like P, Z , etc are written in italic
Line 200: replace citations by references or papers or studies
Line 220: check formatting of (Table S…), something is not right
Line 247: delete redundant space after condition
Line 249: delete redundant space before citations
Line 278: delete redundant space after citation
Line 322: add space before citation
Line 339: check formatting of Lunghi et al 2018c.
Table 1: replace the zero in last column, third row, by a slash (/)
Author Response
# REV 1
Major comment:
Line 307-313: I completely agree with you here, but also think that you should mention another case for a complete record. One common way in comparative biospeleobiology is to contrast a subterranean species to a closely related epigean species. Often these comparisons offer only limited insight as usually species used are not so closely related after all (for example Proteus and Calotriton that you used as a reference here). The other common approach is to contrast subterranean populations with epigean populations in species like Astyanax mexicanus, Asellus aquaticus, Gammarus minus, etc. These are very closely related but also their populations are very distinct. This might be the best approach. Finally, come your model organisms (European cave salamanders and similar), whose subterranean and epigean populations do not differ so markedly as in the previously mentioned species. So, I suggest adding a sentence or two, mentioning the above model organisms and providing appropriate reference. For these I suggest the latest reviews, chapter in the recent Groundwater Ecology and Evolution monograph.
As suggested, we added a short discussion about the cave fish Astyanax mexicanum with these references:
72) Borowsky, R. (2023). Selection maintains the phenotypic divergence of cave and surface fish. American Naturalist, 202: 55-63.
73) Gross, J.B., Boggs, T.E., Rétaux, S., Torres-Paz, J. (2023). Developmental and genetic basis of troglomorphic traits in the teleost fish Astyanax mexicanus, pp. 351-371 in Groundwater Ecology and Evolution (Eds Malard, F., Griebler, C., Rétaux, S.
Minor comments:
Throughout the manuscript genus names are often not written in italic. This is especially apparent in the simple summary, abstract and future directions. Please see to it.
Changed
Line 57: change Amphibians to amphibians
Changed
Line 71: check formatting of citations
Changed
Line 81: replace - after gap by a full stop.
Changed
Line 88: not clear which paper by Lanza you are referring to as there is no year of publication
Added
Line 136: delete redundant space between literature and on
Deleted
Line 137: delete redundant space between with and American
Deleted
Line 193: usually stats labels like P, Z , etc are written in italic
Changed
Line 200: replace citations by references or papers or studies
Line 220: check formatting of (Table S…), something is not right
Changed
Line 247: delete redundant space after condition
Deleted
Line 249: delete redundant space before citations
Deleted
Line 278: delete redundant space after citation
Deleted
Line 322: add space before citation
Added
Line 339: check formatting of Lunghi et al 2018c.
….
Table 1: replace the zero in last column, third row, by a slash (/)
Replaced
Reviewer 2 Report
Comments and Suggestions for Authors
I believe that many aspects of this manuscript have been improved since its previous submission. There are, however, issues that remain to be resolved or improved. In addition, many errors and missing references in the manuscript should have been caught prior to submission – e.g., Line 82 “(but see).” The next version needs to be carefully reviewed before submission.
I have broken down my comments by manuscript section.
Throughout
· The genus name needs to be italicized. There are many instances where Speleomantes is not italicized
· “Parental cares” should be “parental care”
Abstract/ Lay Summary
· Lines 32-33 – The lay summary says that studies were lacking in intraspecific chemical communication - how does that jive with what you have here that says that intraspecific studies were 1/3 of the studies found? I think that this highlights my issue with the intraspecific behavior category (more details below).
· The use of this group as a model species is mentioned in the lay summary, but not in the abstract, which is a bit odd.
· Line 14 – reviewed instead of revised
· Line 16 – directions instead of directs
Intro
· References are cited out of order (e.g., there is no citation to reference 10 between 9 and 11)
· Lines 52-56 – Missing references to support these examples.
· Lines 75-77 – This statement needs a reference
· Line 78 – This paragraph on plethodontids is quite long. I would suggest breaking off info about Old World salamanders into a separate paragraph starting here.
· Line 80-81 – I am not sure what this phrase is – is it meant to be a complete sentence?
· Line 82 – “(but see)” – reference missing in parentheses
· Lines 83-84 – Reference needed for study mentioned here. Also, syntopy? Is this supposed to be sympatry?
· Line 90 – “(e.g.,)” – reference missing in paretheses
· Lines 94-101 - This paragraph seems lacking/unfinished. It needs examples or something to close out the paragraph.
· Line 101 – References skip from #16 to 19 (Line 105)
Methods
· Lines 181-186 - The motivation behind this categorization should be introduced before this - why did you feel it was necessary to sort papers by observational or experimental? Was it determining the extent to which each type of study was conducted? To better understand the types of studies within each behavioral category? It doesn’t need to be much, but there should be some indication of why you kept track of observational vs. experimental studies
·
Results
· Lines 200-202 – These numbers do not align with those in Figure 2 - according to Figure 2, there were 15 Intraspecific social behavior papers and 13 foraging papers.
· Line 207 - Fig 2 shows the breakdown of different types of behavioral studies by obs and exp, so the ref to that figure doesn't make sense here. It should be referenced in the previous sentence and this sentence should reference Table 1.
Discussion
· This section provides a summary and/or additional thoughts on nearly every behavioral category. The predators and parasites papers, however, are not addressed beyond a mention. These studies should be given equal text time as the other behavioral categories.
· Line 220 – “(Table S…)” – This is incomplete
· Lines 249-253 – I would argue that these studies should be considered for this review with caveats. Although the conditions were not natural, was information on behavior not gained? If the authors are promoting more experimental studies of behavior in this group, which often very much differ from reality, why should this experimental sympatry study not be included in this review?
· Lines 262-263 - Given that this gap is not shown in your results, perhaps that suggests that a different behavior classification system is needed. Right now, “intraspecific social interactions” has more results than many of the other behavioral categories. Should it be divided into parental care vs. other intraspecific interactions to highlight this gap? You could also discuss the lack of pheromonal studies in the discussion.
Future Directions
· The start of this section reads like a continuation of the previous section, rather than a new set of ideas. I would rephrase the start of this section so that it independently focuses on future directions.
· Line 320 – Scientific name should be italicized.
Figure 1
· I am not familiar with N^o as an abbreviation for number – is this common?
Comments on the Quality of English LanguageThis manuscript has many grammatical and word choice errors. Some sections are error-free, while others have grammatical issues that impede understanding. I did not pull out specifics below for the sake of time, but moderate English editing is required.
Author Response
# REV 2
I believe that many aspects of this manuscript have been improved since its previous submission. There are, however, issues that remain to be resolved or improved. In addition, many errors and missing references in the manuscript should have been caught prior to submission – e.g., Line 82 “(but see).” The next version needs to be carefully reviewed before submission.
I have broken down my comments by manuscript section.
Throughout
- The genus name needs to be italicized. There are many instances where Speleomantes is not italicized
We thanks Reviewer 2 for his comments and suggestions. We corrected these typos due to journal’s editing errors.
- “Parental cares” should be “parental care”
Changed
Abstract/ Lay Summary
- Lines 32-33– The lay summary says that studies were lacking in intraspecific chemical communication - how does that jive with what you have here that says that intraspecific studies were 1/3 of the studies found? I think that this highlights my issue with the intraspecific behavior category (more details below).
We added more information about the papers belonging to this category in the results section
- The use of this group as a model species is mentioned in the lay summary, but not in the abstract, which is a bit odd.
……
- Line 14 – reviewed instead of revised
Changed
- Line 16 – directions instead of directs
Changed
Intro
- References are cited out of order (e.g., there is no citation to reference 10 between 9 and 11)
Changed
- Lines 52-56 – Missing references to support these examples.
Added
- Lines 75-77 – This statement needs a reference
Added
- Line 78 – This paragraph on plethodontids is quite long. I would suggest breaking off info about Old World salamanders into a separate paragraph starting here.
Changed
- Line 80-81 – I am not sure what this phrase is – is it meant to be a complete sentence?
Changed
- Lines 83-84 – Reference needed for study mentioned here. Also, syntopy? Is this supposed to be sympatry?
Added
- Lines 94-101 - This paragraph seems lacking/unfinished. It needs examples or something to close out the paragraph.
Rephrased
Methods
- Lines 181-186 - The motivation behind this categorization should be introduced before this - why did you feel it was necessary to sort papers by observational or experimental? Was it determining the extent to which each type of study was conducted? To better understand the types of studies within each behavioral category? It doesn’t need to be much, but there should be some indication of why you kept track of observational vs. experimental studies
Information added
Results
- Lines 200-202 – These numbers do not align with those in Figure 2 - according to Figure 2, there were 15 Intraspecific social behavior papers and 13 foraging papers.
We corrected the typo in Fig. 2.
- Line 207 - Fig 2 shows the breakdown of different types of behavioral studies by obs and exp, so the ref to that figure doesn't make sense here. It should be referenced in the previous sentence and this sentence should reference Table 1.
Changed
Discussion
- This section provides a summary and/or additional thoughts on nearly every behavioral category. The predators and parasites papers, however, are not addressed beyond a mention. These studies should be given equal text time as the other behavioral categories.
We deepened the discussion regardin this topic.
- Line 220 – “(Table S…)” – This is incomplete
Changed
- Lines 249-253 – I would argue that these studies should be considered for this review with caveats. Although the conditions were not natural, was information on behavior not gained? If the authors are promoting more experimental studies of behavior in this group, which often very much differ from reality, why should this experimental sympatry study not be included in this review?
This paper is from 1999 and therefore was already included and discussed in the monograph of Lanza et al. (2006). In addition, this was a artificial setting in which one species was introduced in the natural habitat of the other one, and we do not support this experimental approach.
- Lines 262-263 - Given that this gap is not shown in your results, perhaps that suggests that a different behavior classification system is needed. Right now, “intraspecific social interactions” has more results than many of the other behavioral categories. Should it be divided into parental care vs. other intraspecific interactions to highlight this gap? You could also discuss the lack of pheromonal studies in the discussion.
We added this information in the results section.
Future Directions
- The start of this section reads like a continuation of the previous section, rather than a new set of ideas. I would rephrase the start of this section so that it independently focuses on future directions.
We added the new section “Future directions”, following the previous suggestions of Reviewer 1 and Animals formatting guidelines.
- Line 320 – Scientific name should be italicized.
Changed
Figure 1
- I am not familiar with N^o as an abbreviation for number – is this common?
Changed